

# Untangling the Waves: Decomposing Extreme Sea Levels in a non-tidal basin, the Baltic Sea

Marvin Lorenz[1,2], Katri Viigand[3], and Ulf Gräwe[1]

[1]Leibniz Institute for Baltic Sea Research Warnemünde, Rostock, Germany
[2]Federal Waterways Engineering and Research Institute, Hamburg, Germany
[3]Department of Cybernetics, School of Science, Tallinn University of Technology, Tallinn, Estonia

**Correspondence:** Marvin Lorenz (marvin.lorenz@io-warnemuende.de)

**Abstract.** Extreme sea level (ESL) events are typically caused by the combination of various long surface waves, such as storm surges and high tides. In the non-tidal, semi-enclosed Baltic Sea, however, ESL dynamics differ. Key contributors include the Baltic's variable filling state (preconditioning) due to limited water exchange with the North Sea and inertial surface waves, known as seiches, which are triggered by wind, atmospheric pressure, and basin bathymetry. This study decomposes ESL

events in the Baltic Sea into three key components: the filling state, seiches, and storm surges. Our results show that storm surges dominate the western Baltic, while the filling state is more influential in the central and northern regions. Using a numerical hydrodynamic model, we further decompose these components based on their driving forces: wind, atmospheric pressure, North Atlantic sea level, baroclinicity, and sea ice. Wind and atmospheric pressure are the primary forces across all components, with the Atlantic sea level contributing up to 10% to the filling state. These findings offer a deeper understanding

of ESL formation in the Baltic Sea, providing critical insights for coastal flood risk assessment in this unique region.

## 1 Introduction

The threat of extreme sea level (ESL) events escalates along the world's coastlines due to the rising mean sea level. This phenomenon is a leading cause of flooding in coastal areas, affecting more than 100,000 people annually in Europe alone. Moreover, this number is projected to rise in the future (Vousdoukas et al., 2020). Due to anthropogenic climate change,

ESLs are projected to increase not only due to the rising mean sea level (e.g. Fox-Kemper et al., 2021), but also due to the intensification of tropical storms (e.g. Knutson et al., 2010) and changing sea level dynamics due to sea level rise (e.g. Moftakhari et al., 2024), such as increasing tidal amplitudes and surges in estuaries (e.g. McGranahan et al., 2007; Talke et al., 2021) or coastal lagoons (e.g. Bilskie et al., 2014; Lorenz et al., 2023).

ESLs are often a combination of several processes, each leading to an increase in coastal sea level, e.g. high tides, storm

surges, wave setup due to breaking wind waves, river discharge, and other surface waves. Storm surges occurring at high tide are the most common cause of ESLs for most coastal areas. For estuaries, where rivers flow into the sea, the coincidence of a storm surge at high tide with extreme river discharge poses a major flood hazard (e.g. Wahl et al., 2015; Hendry et al., 2019). Generally, when two or more extreme events co-occur, they are called compound events. Since many different components can contribute to ESLs, it is desirable to understand and quantify each to assess each contributor's risk and statistics. However,





decomposing ESLs is challenging. Using tide gauges alone, only a temporal decomposition is possible, e.g. the separation of tides and surges. However, non-linear interactions between the two complicate the separation (Arns et al., 2020) and are, as a result, often neglected in impact studies (e.g. Rueda et al., 2017; Vousdoukas et al., 2018; Kirezci et al., 2020). Numerical models can allow the decomposition of ESL events into their different contributions, as individual processes and forcings can be switched on or off (Parker et al., 2023). In this way, each contributor can be quantified. Numerical models rely on

good input data, such as atmospheric forcing with a good representation of storm systems, to capture storm surges and, hence ESLs correctly. However, long-term, high quality and high resolution input data are often unavailable. Therefore, models are not always able to reproduce every individual event accurately. Nevertheless, comparison with tide gauge data has shown that models can still correctly reproduce ESL statistics. Therefore, models are invaluable for decomposing ESL events into different components and forcings. Most decomposition studies assume a linear superposition of the individual sea level contributors

(Parker et al., 2023). Interactions between the different contributors are assumed to be of second order.

    In this study, we follow the decomposition approach to decompose the ESLs within a non-tidal, semi-enclosed marginal sea, the Baltic Sea (Fig. 1). The Baltic Sea, located in northern Europe, has a mean depth of approximately 55 m. The region experiences postglacial isostatic land uplift of about 10 mm/year in the northern part of the sea, while the very southern region experiences subsidence of about 0.05 mm/year (Vestøl et al., 2019). Long-term global eustatic sea level rise in the Baltic Sea

is not uniformly distributed across the region. It varies from 2 mm/year in the western part of the sea to more than 5 mm/year in the Gulf of Bothnia. About 75% of the basin-averaged sea level rise can be attributed to the external sea level signal while intensifying winds and poleward shifting low-pressure systems explain the spatial variation in the mean sea level trends (Gräwe et al., 2019). The prevailing winds are westerly to southerly (Bierstedt et al., 2015) and ESLs usually occur in autumn and winter when storm systems pass over the Baltic Sea region. A series of clustered storm systems can cause exceptionally

high ESLs in the northern Baltic Sea due to the overlapping effects of previous storms (Rantanen et al., 2024). In the Western Baltic Sea, very high sea levels have been observed in recent years (Groll et al., 2024), including the highest sea levels since 1872 in October 2023 (Kiesel et al., 2024). Events like these cause flooding, which will become more severe with sea level rise (Kiesel et al., 2023b) and pose problems for current flood protection in the future (Kiesel et al., 2023a).

    A recent review of the sea level dynamics and ESL events in the Baltic Sea has been compiled by Weisse et al. (2021) and

Rutgersson et al. (2022) as part of the Baltic Earth Assessment Reports (BEARs). The sea level dynamics and the ESLs of the Baltic Sea have some unique contributions that distinguish the dynamics of the Baltic Sea from other coasts around the globe. First, the Baltic Sea has negligible tides (e.g. Gräwe and Burchard, 2012). In addition, since the Baltic Sea is separated from the North Sea by the shallow and narrow Danish Straits, water exchange between the Baltic Sea basin and the North Sea is hindered. The Danish Straits effectively act as a low-pass filter, filtering out high-frequency surface waves, such as storm surges

and tides from the North Sea. On the other hand, the slow water exchange on a time scale of a week and longer determines the mean sea level of the Baltic Sea, also called the *filling* state or, in the context of ESL events, the *preconditioning* (Leppäranta and Myrberg, 2009; Madsen et al., 2015). This exchange is mainly controlled by the westerly winds (Gräwe et al., 2019). Similarly, local preconditioning in sub-basins can contribute to ESLs, e.g. up to 1 m in the Gulf of Riga (Männikus et al., 2019). Although the Baltic Sea has negligible tides, there are basin-wide inertial surface waves with fixed frequencies between





$(13\,\mathrm{h})^{-1}$ and $(44\,\mathrm{h})^{-1}$ which are given by the basin geometry (e.g. Wübber and Krauss, 1979; Zakharchuk et al., 2021). These inertial waves are called *seiches* and are excited by perturbations in the sea level. Their amplitudes can be as large as 17 cm, although their initial perturbations were smaller (Zakharchuk et al., 2021). Together with *storm surges*, these three temporal components are the main contributors to ESL events. At the coast, local sea levels may be directly elevated by wind-driven wave effects, such as wave setup (Su et al., 2024) or wave run-up. Meteotsunamis can be an additional component (Pellikka

et al., 2020, 2022). Due to local geometries and orientations of the coastal topography, different regions are susceptible to different combinations of compounding contributions. Therefore, the return levels of ESLs are very heterogeneous. The return levels for a 30-year ESL event range from 0.7 to 2.5 m above mean sea level (Lorenz and Gräwe, 2023a), with the highest ESLs occurring at the head of the Gulf of Finland, the Gulf of Riga, and in the Western Baltic Sea (Wolski et al., 2014; Wolski and Wiśniewski, 2020). The historical storm surge of 1872 reached sea levels of more than 3 m in the Western Baltic Sea (e.g.

Hofstede and Hamann, 2022), with an estimated return period of about 1,000 to 3,000 years, depending on the location and the statistics (MacPherson et al., 2023). This event resulted from a very high preconditioning and an exceptionally high storm surge, which could have been even higher (Andrée et al., 2023). Another event was recorded in 2005 for the Gulf of Riga with a maximum sea level of 2.75 m at Pärnu (Suursaar and Sooäär, 2007; Mäll et al., 2017). Historical events like these are often considered statistical outliers that do not seem to belong to the main population of extreme events (Hofstede and Hamann,

2022; Suursaar and Sooäär, 2007). However, their inclusion in the statistics is beneficial as they can significantly change the design sea level for coastal protection (MacPherson et al., 2023).

Storm surges are driven by winds and air pressure, the primary forces acting on the sea surface. However, the enclosed nature of the Baltic Sea limits the distance over which the momentum for storm surges can be deposited. Additionally, the northern Baltic Sea is partially covered by sea ice from winter to spring (e.g. Luomaranta et al., 2014). Sea ice acts as a lid, hindering

the transfer of momentum from the wind to the water, reducing the ESLs by several decimetres (Zhang and Leppäranta, 1995). These factors, combined with short fetch lengths, generally result in lower return levels for the Gulf of Bothnia, except at its northern head, where the fetch length is maximal for southerly winds (Wolski et al., 2014; Lorenz and Gräwe, 2023a).

Among all the contributors to ESLs in the Baltic Sea, this study quantifies the contributions of the filling or preconditioning, seiches, and the storm surges to ESL events by performing a temporal decomposition. The second focus is quantifying different

forcings' relative importance on the ESL events by performing numerical simulations. We quantify the role of the following forcings: wind, air pressure, North Atlantic sea level, baroclinicity, and sea ice.

## 2 Data and Methods

### 2.1 Observational gauge data

The observational tide gauge data are obtained from the European Marine Observation and Data Network (EMODnet, https:

//emodnet.ec.europa.eu, last access: 5 June 2021) and the Global Extreme Sea Level Analysis (GESLA Woodworth et al., 2016; Haigh et al., 2022). Both data sources provide quality-controlled sea level data at hourly frequency and high accuracy (1 cm). We have selected 70 gauges (see Fig. 1 for their locations) to i) study the temporal decomposition of the observed ESL events





**Table 1.** Overview of the different simulations performed in this study.

| Name | Description |
|---|---|
| Full | Simulation where all components are included. |
| NoWind | Simulation where the wind speed is set to zero, $u_{10} = v_{10} = 0$ |
| NoPresgrad | Simulation where the atmospheric pressure is constant, $P = 1000\,\text{hPa}$. |
| NoAtl | Simulation where the sea surface elevation, i.e. the sea level at the open boundary is set to zero. |
| TSClim | Simulation where the inter-annual variability of temperature of salinity are not included |
| IceClim | Simulation where the inter-annual inter-annual variability of sea ice cover is not considered. |
| NoIce | Simulation where the sea ice cover is not considered. |

and ii) validate the modelled ESL events to ensure that the temporal decomposition agrees with the observations. Before any analysis, the long-term linear trend of the mean sea level was subtracted for each tide gauge. De-trending removes mean sea
level rise and glacial isostatic adjustments (GIA Peltier, 2004) from the time series.

## 2.2 Numerical Simulations

We use the General Estuarine Transport Model (GETM, version 2.5 Burchard and Bolding, 2002) to simulate the sea level dynamics of the Baltic Sea. The model configuration is the same as in Lorenz and Gräwe (2023a) for their ensemble member *UERRA (baroclinic)*. The configuration uses a model chain starting in the Northwest Atlantic (5 NM resolution, bottom rough-
ness $z_0 = 5\,\text{mm}$) to generate boundary conditions for the North Sea and Baltic Sea domain (1 NM resolution, bottom roughness $z_0 = 1\,\text{mm}$, Fig. 1), see e.g. Gräwe et al. (2015). Along the boundary of the Northwest Atlantic domain, air pressure-induced sea level changes (inverse barometric effect) are prescribed by the ERA5 reanalysis (Hersbach et al., 2020), which is also used as the atmospheric forcing for this setup. In this way, large pressure systems over the Atlantic are included in the model chain. The North Sea and Baltic Sea setup is forced by the *Uncertainties in Ensembles of Regional ReAnalyses* (UERRA Ridal et al.,
2017), where the winds are increased by 7% to better represent the ESLs in the Western Baltic Sea (Lorenz and Gräwe, 2023a). Tides are not considered as they are negligible for the Baltic Sea. Static monthly mean density fields and sea ice cover are obtained from Gräwe et al. (2019). Wind stress reduction by sea ice is considered as in Gräwe et al. (2019). The advection scheme is the Superbee TVD scheme (Pietrzak, 1998).

We exclude different forcing components in distinguished model simulations to study the effects of different forcings on
the sea level, see Tab. 1. The respective difference to the *Full* simulation, where all components are included (the same model simulation as used by Lorenz and Gräwe (2023a)), is then attributed to be the effect of the excluded forcing. A detailed description of the decomposition approach is presented in the following sections.

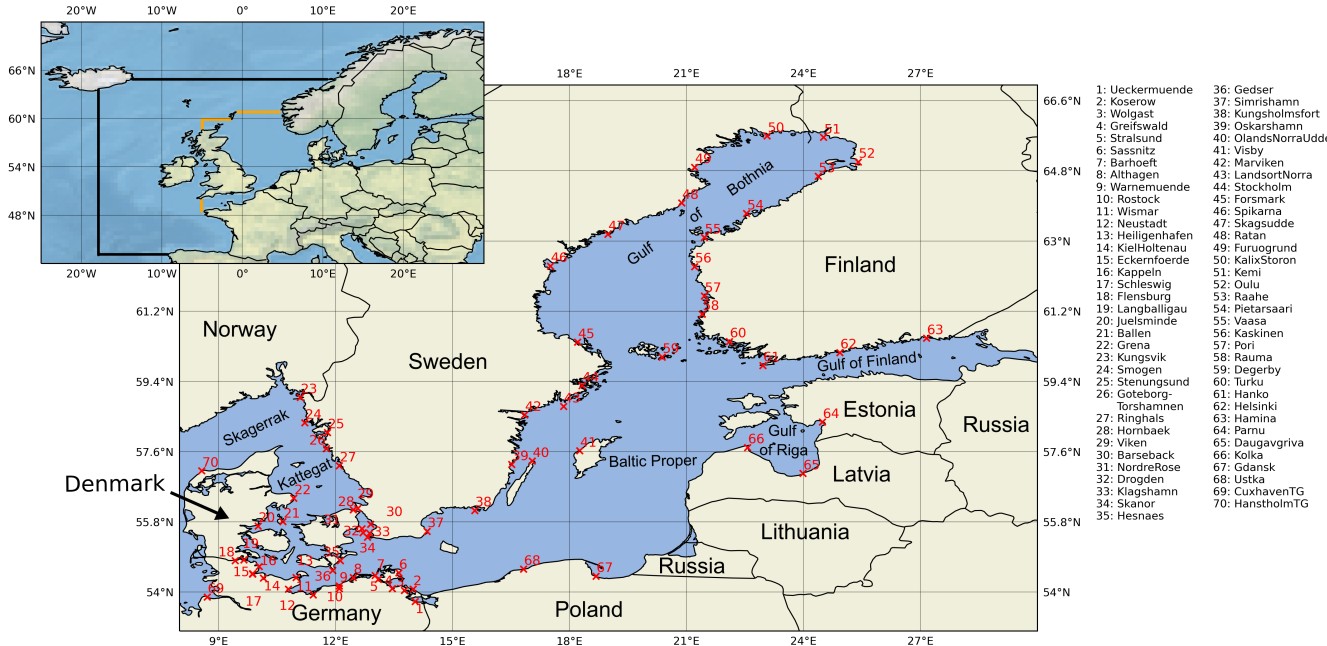

**Figure 1.** Model domain and station locations (red crosses and numbers) used for the temporal decomposition and model validation. The black line indicates the boundary of the coarse Northwest Atlantic Ocean domain. The orange lines mark the boundaries of the one nautical mile domain of the North Sea & Baltic Sea.

## 2.3 Decomposition Approach

### 2.3.1 Temporal decomposition

We first split the time series into three contributions: the surge $\eta_{\mathrm{surge}}$, the filling state of the Baltic Sea $\eta_{\mathrm{fill}}$, and the seiches $\eta_{\mathrm{seiche}}$:

$$\eta_{\mathrm{ESL}} = \eta_{\mathrm{surge}} + \eta_{\mathrm{fill}} + \eta_{\mathrm{seiche}}. \tag{1}$$

The filling state $\eta_{\mathrm{fill}}$ is computed by applying a Butterworth low-pass filter with a cut-off frequency of 7 days, corresponding to the weekly timescale described by Soomere and Pindsoo (2016) and Pindsoo and Soomere (2020). To extract the seiche

contribution $\eta_{\mathrm{seiche}}$ from the time series, we consider a time window of $\pm 7$ days around the peak sea level and force a fit of fixed frequency oscillations $f_i$:

$$\eta_{\mathrm{seiche}} = \sum_i a_i \sin\left(\omega_i t - \phi_i\right), \tag{2}$$

where $i$ denotes the index of the frequencies considered, $a_i$ and $\phi_i$ are the fitted amplitude and phase lag of the seiche, respectively, and $\omega_i = 2\pi f$. For the Baltic Sea, the dominant frequencies $f_i$ we consider are: $(13\,\mathrm{h})^{-1}, (23\,\mathrm{h})^{-1}, (25\,\mathrm{h})^{-1}, (27\,\mathrm{h})^{-1}, (29\,\mathrm{h})^{-1},$



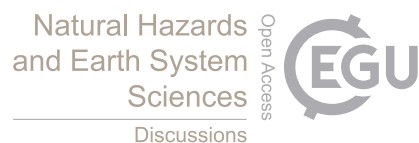
$(41\,\mathrm{h})^{-1}$ (Wübber and Krauss, 1979; Zakharchuk et al., 2021). Note that even more frequencies could have been added. However, the errors introduced by neglecting these additional frequencies are expected to be small, as we consider $\mathcal{O}(100)$ events per station. The surge component is then computed with:

$$\eta_{\mathrm{surge}} = \eta_{\mathrm{ESL}} - \eta_{\mathrm{fill}} - \eta_{\mathrm{seiche}}. \tag{3}$$

### 2.3.2 Decomposition into forcing contributions

In addition to the temporal decomposition, we assume that each contributor in (1) can be linearly decomposed as the sum of the sea levels driven by the forcing components, namely wind $\eta_{\mathrm{wind}}$, air pressure $\eta_{\mathrm{pres}}$, sea level of the Atlantic $\eta_{\mathrm{Atl}}$, inter-annual variability of the baroclinic velocities due to seawater density gradients $\eta_{\mathrm{barocl}}$, sea ice $\eta_{\mathrm{ice}}$, and the inter-annual variability of the sea ice cover $\eta_{\mathrm{icevariability}}$:

$$\eta_j = \eta_{j,\mathrm{wind}} + \eta_{j,\mathrm{pres}} + \eta_{j,\mathrm{Atl}} + \eta_{j,\mathrm{barocl}} + \eta_{j,\mathrm{ice}} + \eta_{j,\mathrm{icevariability}} + \eta_{j,\mathrm{res}}, \tag{4}$$

where $j$ denotes *surge*, *fill*, and *seiche*, respectively. Since there are interactions between the sea levels driven by the different contributors, we have included the residual term $\eta_{\mathrm{res}}$, which essentially represents the difference between $\eta_j$ and the sum of the sea level contributions on the right-hand side, excluding the residual term itself. To extract the effect of each forcing, we subtract the respective simulation where we have excluded one component from the simulation *Full* (compare Tab. 1). For example, the time series of the sea level evolution due to wind forcing is computed by:

$$\eta_{\mathrm{wind}} = \eta_{\mathrm{Full}} - \eta_{\mathrm{NoWind}}. \tag{5}$$

The other forcing contributions are computed accordingly.

### 2.4 Work flow of the time series analysis

As a first step, the long-term changes in the mean sea level are excluded from the time series by de-trending; see Fig. 2a. The filling time series $\eta_{\mathrm{fill}}$ is then computed by applying the Butterworth low-pass filter with a cut-off frequency of 7 days, as
mentioned previously. ESL events are identified in the de-trended time series by applying a peak-over-threshold method. The threshold is determined by the 99.7th percentile of the time series, and we only consider events that are separated by more than 48 hours (Arns et al., 2013). For the modelled time series spanning 58 years from 1961 to 2018, approximately 80 to 250 events are identified per station, which are shown in Fig. 1, see Fig. 2a for the identified events. The number of events in the observed time series may differ due to the varying length of the time series and data gaps. Note that the search for ESL events
is only performed for the *Full* simulation.

We consider a time slice of plus-minus 7 days for each event around the peak sea level. From the resulting shorter time series, the respective filling is first subtracted. Then, the seiches are forced-fitted with eq. (2), followed by the computation of the surge component with eq. 3. An exemplary temporal decomposition of the observed Warnemuende time series (1, station 9) is shown in Fig. 2b. For each temporal component, we record the respective sea level occurring at the time of the peak sea



level (dashed line in Fig. 2b). This results in a sample of sea levels for the total peak level and its respective components from
the temporal decomposition. In addition, the maximum filling and maximum seiche level values $\pm 24\,\mathrm{h}$ within the ESL peaks
are stored to assess the potential ESL if all three components were to reach their peaks simultaneously.

The components are normalised to the peak level for each event and stored in a histogram with a bin size of 0.01, ranging from
-0.5 to 1.2. Each histogram is used to fit a Gaussian distribution to extract the mean and standard deviation; see Fig. 2d. Similar
to the methodology described, each temporal component of *filling*, *seiche*, and *surge* is decomposed into the contributions of
the different forcings, as shown in Fig. 2c,e for the *surge* component as an example. Since the sea ice component can only lower
the sea level, the Gaussian fit approach unsuitable, due to its non-Gaussian distribution. Therefore, for the sea ice component,
the mean and standard deviation are computed directly from the sea levels and then normalised.

Note that the distribution of ESLs generally follows a tail-end statistical distribution such as the Generalised Extreme Value
or Generalised Pareto distribution. The filling component follows a classical asymmetric quasi-Gaussian distribution (Soomere
et al., 2015). However, as we generate samples of relative contributions to normalised peak sea levels, these samples are
expected to follow a Gaussian distribution (except for the sea ice component).

## 3  Results

### 3.1  Temporal decomposition

The temporal decomposition indicates that in the Western Baltic Sea, the surge component accounts for more than 60% of the
ESL events for all stations (Fig. 3a,b). Following the surge component, the seiche and filling components contribute almost
equally. For the westernmost stations, the seiche contribution exceeds that of the filling. In contrast, stations located along
the eastern German coast show a greater contribution of the filling component, indicating that these stations are closer to the
seiches' pivot point(s) and, therefore experience smaller seiche amplitudes. Note that the seiche periods of $23\,\mathrm{to}\,29\mathrm{h}$ are in
the range of the duration of the surge events in this region of the Baltic Sea (Kiesel et al., 2023b). Along the southwestern
coast of the Baltic Sea towards Sweden, the largest contribution shifts from the surge to the filling component. The filling
component contributes 50 to 75% of the total ESL along the coasts of the Baltic Proper and the Gulf of Bothnia (Fig. 3a,c). For
most of the Swedish coast, the remaining contribution to the ESL is the surge component, with almost negligible seiches (Fig.
3a,d). A similar pattern is found on the Finnish coast in the Gulf of Bothnia. In the Gulf of Finland, the surge is the second
largest contributor, followed by the seiche component, with the filling component remaining the largest contributor. The surge
component becomes more significant as a tide gauge is located further east in the Gulf of Finland, due to the increased fetch
length over which the wind can transfer momentum to the sea. In the Gulf of Riga, the seiche component is again negligible
and the contribution of surges to the ESLs is nearly as significant as that of the filling. For the Polish stations, Gdansk and
Ustka, the decomposition is similar to that of the Gulf of Riga, with almost negligible seiches.

Comparison of the gauge data decomposition with the numerical model results, Fig. 3b-d, shows that the model statistics
agree with the observed statistics. Differences are found for gauges that are located within coastal lagoons or estuaries, such
as Ueckermuende, Althagen, Barhoeft, Kappeln, Schleswig, and Gdansk, since the hydrodynamics are not resolved at the 1




**Figure 2.** Exemplary results of the workflow for the decomposition of ESL events: a) The de-trended time series of the station *Warnemuende*. The orange crosses indicate all considered ESL events. The black arrow denotes the exemplary event that is decomposed in panels b) and c). b) Temporal decomposition into *filling*, *seiche*, and *surge* components around the peak sea level of the event. c) Decomposition of the *surge* component into the different forcings. The dashed black line in b) and c) marks the time of the peak sea level. d) The histograms and fitted Gaussian distributions of the temporal decomposition using all the events marked in a). e) Similar to d) but for the decomposition into forcing contributions for the *surge* component.





N.M. resolution, as discussed by Lorenz and Gräwe (2023a). Note that even a 200 m resolution is insufficient to reproduce the hydrodynamics in these areas, resulting in the model overestimating ESLs in these regions (Kiesel et al., 2023b).

Although located outside of the semi-enclosed Baltic Sea, stations in the Kattegat and in the North Sea, still exhibit large filling contributions of 40-50%. This shows that, at least for ESL events, low-frequency waves contribute to the slow sea level variability. Note that tides are not included in the simulations, so the low-frequency variability is not a superposition effect such as the spring-neap cycle. This further indicates that the low-frequency filling also significantly contributes to ESLs in the more open eastern North Sea. The model deviates from the observations for the Cuxhaven station in the southeastern North

Sea because tides are not included in the model.

### 3.1.1    Correlations between the components

A correlation analysis of the three temporal components (Fig. 4a-c) reveals that the filling and the surge components are negatively correlated for the western and central Baltic Sea (Fig. 4a). The filling and the seiche components show only a weak negative correlation (Fig. 4b). The seiche and surge components are positively correlated for the central and eastern Baltic Sea

(Fig. 4c).

The negative correlation between surge and filling is surprising at first sight, as it is known that most ESLs in the Baltic Sea are a coincidence of both contributions. However, the negative correlation between the filling and surge components is partly an artefact of the decomposition method: Since the peak sea level of each event is fixed, a particularly high surge would naturally coincide with a lower filling state relative to the mean of the Gaussian distribution. Otherwise, the average relative

ESL would be higher than 100% and thus the fitted Gaussian distributions would be wrong. The same applies to a low surge and a high filling state. Therefore, these two components must be negatively correlated in this decomposition approach of the mean ESL. For the Warnemuende station this negative correlation is clearly visible, both for the modelled and the observed ESL events (Fig. 5, here with non-normalised sea levels). Of course, events where both the filling and the surge components are very high relative to their mean are also possible and observed (Groll et al., 2024). For example, some of the highest peaks

at Warnemuende, exceeding 1.5 m (Fig. 5b, events numbers 33, 52, 73, 110), are formed by the combination of very high surge and high filling components. However, the probability of occurrence of extremely high filling and surge combined ESL events is low. Nonetheless, these events are often among the most devastating ones and can lead to dangerous inundations in the coastal regions, e.g. the historic 1872 surge (Hofstede and Hamann, 2022).

The positive correlation between the seiche and the surge can be explained by the initialisation of the seiche by preceding

weather systems before the actual surge occurs. The stronger the subsequent system, the higher the seiches initialised by the previous system are likely to be. It is also likely that the frequency of the seiches is comparable to the duration of the surges, so the forced fit may overestimate the height of the seiches during the ESL event.

### 3.1.2    Potential increases of maximum sea levels by time shifting the components

Since the low-frequency waves of the filling component and the high-frequency contributions from seiches and surges operate

on different time scales, the timing of their peaks should be independent of the filling levels, essentially making them random.





**Figure 3.** Temporal decomposition of the peak sea levels: a) Map of the temporal decomposition of the ESL events in the observations for each station. b) Comparison of the modelled *surge* component with the *surge* component from the observed sea level data. The mean (dots) and standard deviation (caps) are obtained from the Gaussian distribution fits (except for sea ice simulations, see Section Workflow), compare Fig. 2d. c) As b) but for the *filling* component. d) as b) but for the *seiche* component.

**Figure 4.** Correlation maps for the three temporal components: a) Correlation coefficient for *surge* and *filling* components. b) Correlation coefficient for *filling* and *seiche* components. c) Correlation coefficient for *seiche* and *surge* components.


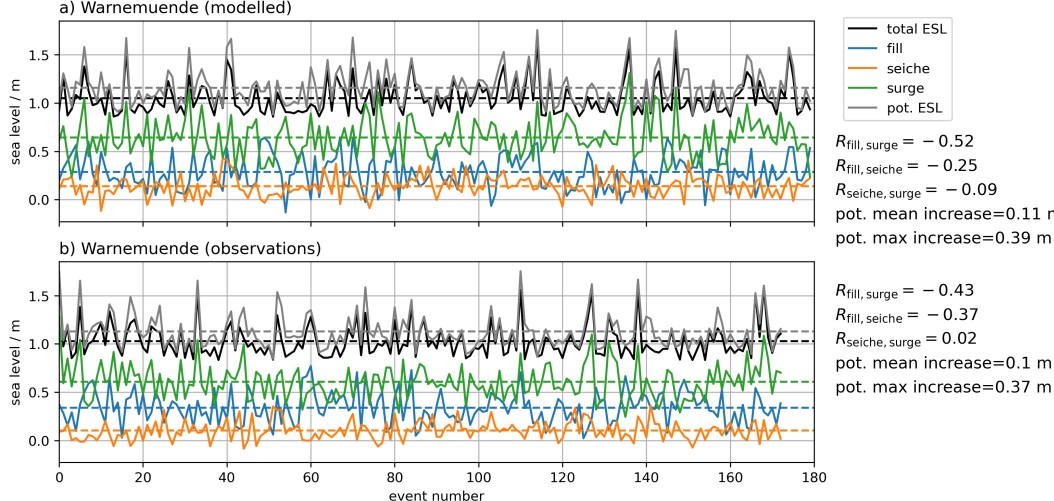

**Figure 5.** Illustration of the temporal decomposition and the correlation between the components for the station Warnemuende. a) The three components *filling*, *surge* and *seiche* for the modelled ESL event. b) As a) but using the observed ESL events. In grey, the maximum ESL heights are shown, e.g. the potential sea level if the surge peak, the seiche peak, and the maximum filling (all $\pm 24\,$h around surge peak) co-occurred.

Therefore, we assess the potential increase in peak sea levels under a worst-case scenario, where the peaks of all three components align. Specifically, we examine the maximum filling levels and seiche heights within a 48-hour window surrounding the ESL peaks. For the Warnemuende station, the mean and maximum potential increases due to these compounding components are $11\,(10)\,$cm and $39\,(37)\,$cm respectively, where the values in the brackets are from the observations (grey in Fig. 5). A more

detailed examination of these potential increases shows that the filling component has less potential to increase the ESLs than the seiche component (Fig. 6a-d). On average, the filling could have increased the ESLs by only a few centimetres (Fig. 6a), whereas the maximum increase could have been in the range of 10-20 centimetres (Fig. 6b). If the seiche peaks were aligned with the surge peaks, the mean increase could reach $10\,$cm (Fig. 6c) and the maximum increase of up to 30-40 cm (Fig. 6d). The largest mean increases are found in the western Baltic Sea and at the head of the Gulf of Finland with values between

10-15 cm (Fig. 6e). The largest maximum increases are also found in these regions with values reaching up to $50\,$cm (Fig. 6f). The large potential increases in the northern Baltic Sea most likely correspond to the effect of seiches introduced by preceding storms, i.e. the clustering of passing storm systems as described by Rantanen et al. (2024).

### 3.2    Decomposition into forcing components

As expected, the surge component is almost entirely explained by the winds. Throughout the Baltic Sea, winds account for

80% and more of the surge heights (Fig. 7). The remainder is mainly explained by atmospheric pressure. Winds continue being the most important surge forcing for the Kattegat, north of the Danish Straits and outside the Baltic Sea basin. However,



**Figure 6.** Potential increases of the ESL event if the three temporal components were compounding, e.g. their maxima occurred simultaneously: a) Mean potential increase from the filling component (difference maximum filling around the ESL peak ($\pm 24$ h) to the actual filling at the ESL event). b) Maximum increase of the filling component. c) As a) but for the seiche component. d) As b) but for the seiche component. e) Potential mean sea level increase if the surge peak, the seiche peak, and the maximum filling occurred at the same time (sum of a) and c)). f) As e) but for the maximum ESL increase within all ESL events in the respective grid cell (sum of b) and d)).





the Atlantic sea level contributes up to 30%. This indicates that some of the surges in this region originate outside the model domain. Their fetch lengths exceed the distance from the Kattegat/Skagerrak to the open boundary. Therefore, the origin of these surges is introduced by the boundary conditions of the Northwest Atlantic through the inverse barometric effect. The winds within the model domain then amplify these surges. The effect of sea ice is limited to the Gulf of Bothnia and the head of the Gulf of Finland, where the surge height can be reduced by 10% on average and even more for individual events, which is in the order of the expected magnitude (Zhang and Leppäranta, 1995).

Similarly, winds account for the seiche component almost everywhere, with minor contributions from atmospheric pressure in the Baltic Sea (Fig. 8). The Atlantic component also plays a role for Kattegat near the Danish Straits. Generally, the fitted seiche periods are based on the seiche frequencies in the Baltic Sea basin. Therefore, the chosen seiche frequencies likely do not correspond to the actual seiches for this region. This discrepancy may explain why the seiche component in the Kattegat and Skagerrak is the smallest contributor to the ESLs. Furthermore, sea ice has a minor influence on the seiches in the Gulf of Bothnia.

The filling component is also mainly determined by the wind forcing (Fig. 9), with contributions ranging from 50% in the Kattegat and Skagerrak to more than 80% in the Western Baltic. For the Western Baltic Sea, atmospheric pressure is the second largest contributor to the filling component. The other forcings contribute a few percent to the mean filling for the rest of the Baltic Sea basin. In contrast to the surge and the seiche components, the residual term shows values of up to 20% within the Baltic Sea. The wind and Atlantic sea level forcings explain most of the filling for the Skagerrak and Kattegat. However, the residual term shows values up to 40% across the Baltic Sea. These large residuals indicate that the assumed linearity of the decomposition into forcings does not hold well for the filling component, likely due to temporal shifts of low-frequency waves.

The results of the forcing decomposition show that the inter-annual variability of sea ice has a negligible effect on ESLs due to its minimal contribution to any of the three temporal components: surge, seiche or filling (Figs. 7-9). The inter-annual variability of the baroclinicity has no effect on the high-frequency surge and seiche components (Figs. 7- 8) and only a minimal effect on the filling component. The Atlantic sea level mainly affects the surge component for the Kattegat and Skagerrak, while its contribution to the filling component is limited to a few percent. As expected, the wind and air pressure forcing are the main forcings for all three components, with the wind being the most important.

## 4 Discussion

### 4.1 Components

#### 4.1.1 Importance of the components and forcings

Our results show two distinct temporal compositions of the ESLs: In the Western Baltic Sea, the primary component is the surge, which acts upon the local filling state and is modulated by the seiche component. In contrast, filling is the primary driver in the rest of the Baltic Sea basin, with relatively smaller surges acting on top. Again, the seiches modulate the peak sea level. For the Kattegat and Skagerrak outside the semi-enclosed basin, the contributions of filling and surge are almost equal, with



**Figure 7.** Decomposition of the *surge* component into forcing contributions: a) Pie charts of the relative forcing contribution for each station scattered over the Baltic Sea. b) Relative contributions regarding mean and standard deviation (based on Gaussian fits) for each station.

**Figure 8.** Decomposition of the *seiche* component into forcing contributions: a) Pie charts of the relative forcing contribution for each station scattered around the Baltic Sea. b) Relative contributions regarding mean and standard deviation (based on Gaussian fits) for each station.

**Figure 9.** Decomposition of the *filling* component into forcing contributions: a) Pie charts of the relative forcing contribution for each station scattered around the Baltic Sea. b) Relative contributions regarding mean and standard deviation (based on Gaussian fits) for each station.





the seiche being the least contributing component. Different primary contributors can lead to variations in the characteristics

of ESLs, potentially causing differences in their dynamics, persistence, and probability of occurrence at specific locations.

This distribution of two distinct contributors to ESLs corresponds to the dominant wind systems in the Baltic Sea region. Westerly winds generate high filling (Gräwe et al., 2019) and often cause storm surges on the west-facing coasts of the Baltic Sea. There, the filling is the main contributor, and surges also play a significant role. The east-facing coasts of the Baltic Sea are sheltered from these prevailing winds, so that the filling component the most influential factor there, with the exception

of the Western Baltic Sea. While strong winds from the north and east are less frequent (Soomere and Keevallik, 2001), these winds cause surges along the coasts of Germany, Denmark and Sweden.

Although the importance of the filling component has been recognised in the literature before (e.g. Soomere and Pindsoo, 2016; Pindsoo and Soomere, 2020), this study provides quantitative figures that underline its significance. For the Western Baltic Sea, its mean contribution is less than the contribution of the surge, which is consistent with the recent results of Groll

et al. (2024), who studied the contribution of surge and filling in the Western Baltic Sea. This is partly because water can flow into the Kattegat, Skagerrak and the North Sea on this time scale. Nevertheless, it is a fact that both filling and surge have compounded for the highest ESL events that caused severe historical inundation and damage, e.g. 1872 (Hofstede and Hamann, 2022) or 2005 (Suursaar and Sooäär, 2007; Mäll et al., 2017).

We have demonstrated that temporal shifts towards simultaneous alignment of the maximum sea levels of the three temporal

components can increase the ESLs by several decimetres within the respective event. It has to be noted that the forced-fitting approach can lead to a positive bias of potential increases, especially if the surge duration is similar to one or more of the seiche periods. Therefore, the maximum potential increase due to the seiches (Fig. 6d) and hence the maximum total increases (Fig. 6f) are most likely overestimated. Nevertheless, the potential average increase is in the order of 10-20 cm. These values do not represent the theoretical maxima of ESLs in the respective regions, which would occur if all the components investigated were

to reach their highest observed levels, which is statistically unlikely but possible (e.g. Andrée et al., 2023; Groll et al., 2024).

Our analysis of the forcing contribution clearly shows that wind and air pressure are by far the most important contributors to the ESLs in the Baltic Sea region. This result is expected since most ocean surface waves are forced by momentum transfer from the atmosphere to the ocean by winds or by atmospheric pressure via the inverse barometric effect.

Sea ice is known to reduce the local momentum transfer to the water and therefore reduce storm surges (see Zhang and

Leppäranta, 1995, for the Baltic Sea). For the northern Baltic Sea, sea ice is present every winter during the storm season, which we have included in our simulations. Our model simulations suggest that sea ice reduces the surge component by 10% on average, translating into sea level changes in the range of decimetres shown by Zhang and Leppäranta (1995). In regions where sea ice is present in winter, the filling component is the main contributor to ESLs, which is almost unaffected by sea ice as filling is a basin-wide process. However, with decreasing levels of sea ice due to climate change (e.g. Meier et al., 2022a, b,

and references therein), the contribution of storm surge to ESLs is likely to increase in the future in regions that are currently covered by annual sea ice. In addition, with decreasing ice cover, the average wave loads and annual wave energy flux are expected to increase by about 5% and up to 82% respectively (Najafzadeh et al., 2022). The same arguments can be made for





the inter-annual variability of sea ice. Our results show that this contribution to the ESLs is currently very small. However, in a warming climate, the contribution of inter-annual variability of sea ice may increase.

The inter-annual effects of baroclinic density fields to ESLs are found to be negligible, indicating that baroclinic contributions to water transports are not important for ESL events in the Baltic Sea.

### 4.1.2    Excluded components and forcings

Many coastal regions of the world have tidal regimes that allow ESLs to occur only during high tide and also interact with storm surges (Arns et al., 2020). The tides are generally very small ($< 10$ cm) and confined to the Western Baltic Sea (e.g.

Gräwe and Burchard, 2012). Therefore, we expect tide-surge interactions to be small. Their mean relative contribution to ESL events should be zero, as the occurrence of the surge should be uniformly distributed over the tidal cycle. Consequently, we have deliberately excluded the tidal component in our temporal decomposition. For tidal regions it can easily be added to the decomposition.

    For the decomposition into forcings, we have also omitted the process of wave setup, e.g. momentum deposition due to

breaking surface wind waves, which can also contribute to ESLs (Longuet-Higgins and Stewart, 1964). In the eastern Baltic Sea in the Gulf of Finland, wave setup can contribute several decimetres to ESLs in some coastal segments, resulting in relative contributions of up to 50% (e.g. Soomere et al., 2013). It should be noted, however, that these estimates are subject to large uncertainties due to assumptions made about the relationship between the offshore wave heights and the wave setup itself. In the Western Baltic Sea, the effect of wave setup on ESLs is only a few centimetres (Su et al., 2024). However, the

potential contribution of wave setup can be substantial in specific locations. The challenge in accurately assessing the wave setup component lies in the need for high-resolution data on wind wave properties within the surf zone, which even state-of-the-art hydrodynamic models struggle to capture on large scale, such as the entire Baltic Sea. Running a wave model with such fine resolution at this scale is not feasible. As a result, distinguishing wave setup from the direct momentum transfer caused by wind remains a task for future research.

In general, the exclusion of wave setup should not affect our results much, as the model can reproduce the ESL statistics (Lorenz and Gräwe, 2023a) and the temporal decomposition of the observations (Fig. 3). Thus, the transfer of momentum from the atmosphere to the ocean via the transfer bulk formula is able to reproduce the ESLs. However, it should be noted that the wind speeds were increased by 7%, which was necessary to match the ESLs with the observations, especially in the Western Baltic Sea, see (Lorenz and Gräwe, 2023a) for a discussion of this issue. This increase may partly compensate for the

missing wave setup. Nevertheless, this would not diminish the relative importance of the wind forcing. It would only split the wind component into a direct sea level component, the generation of long surface waves, and an indirect sea level input via the breaking of surface wind waves within the surf zones.

    Similar to the wave setup, we have excluded meteotsunamis (Monserrat et al., 2006; Pattiaratchi and Wijeratne, 2015). These are tsunami-like surface waves generated by the matching of the propagation speeds of a small atmospheric pressure jump and

the induced surface wave, e.g. a Proudman resonance (Proudman, 1929). The reasons for neglecting meteotsunamis are simple: First, the temporal and spatial resolution of the meteorological forcing is not high enough to accurately resolve the propagating





pressure system to generate meteotsunamis in the numerical model. Second, the hourly resolution of the observational data used is also too coarse to resolve meteotsunamis that occur on faster time scales (minutes). Nevertheless, meteotsunamis are a common phenomenon in the Baltic Sea (Pellikka et al., 2020, 2022) with high sea level contributions in the order of decimetres.

We have also ignored the influence of river discharge since our model's coarse resolution does not sufficiently resolve the estuaries and constrictions where river discharge increases sea level.

### 4.1.3 The residual term and non-linear interactions

In decomposing the different forcing components, a residual term has been added to the right hand side of eq. (4) to account for non-linear effects between the forcing contributions and unaccounted forcings. The residual term is not important for the seiche

component. However, for individual events, the assumption of linear superposition may not hold since the standard deviation of the residual term is in the range of 10-20%. For the surge component, the decomposition into forcings works well. Both the mean residual and the standard deviation remain small, except for the Kattegat region. There, the residual term becomes notably large, indicating the presence of non-linear interactions. These interactions could affect both the sea level height and the timing, leading to temporal shifts. Similarly, the residual term for the filling component is non-zero throughout the Baltic Sea.

Overall, the residual term is not negligible for any temporal component, clearly showing the interaction between the different forcings on the simulated sea levels. Nevertheless, our results quantify the relative importance of the different forcings to each other, which should be independent of the residual term.

    The temporal decomposition performed in this study (1) does not allow the quantification of non-linear interactions between the different time scales. Although these non-linear interactions are included in the respective time series, the computation

of the three components automatically attributes all non-linear interactions to the surge component, see eq. (3). However, the non-linear interactions between the temporal contributors are expected to be small in the Baltic Sea (Arns et al., 2020).

### 4.2 Consequences for ESL statistics

The results indicate that the three temporal components are correlated: there is a negative correlation between filling and surge, a weak negative correlation between filling and seiche, and a positive correlation between the surge and seiche. This suggests

that the contributors are not independent, which has implications for classically applied ESL statistics. The observed negative correlation between components hints that return levels based on classical ESL statistics may be overestimated. However, it is important to emphasise that these negative correlations are based on the statistics of all values exceeding 99.7 percentile. Furthermore, as we have already described, the negative correlation can be an artefact of the decomposition approach (section 3.1.1). It is commonly assumed that the ESL distributions can be described by either the Generalised Extreme Value (GEV)

distribution or the Generalised Pareto Distribution (GPD Coles et al., 2001). However, as, for example, Suursaar and Sooäär (2007) note, some outliers do not fit these distributions. Kudryavtseva et al. (2021) suggest that these 'outliers' could be caused by a sequence of storms that initially increase the filling state of the Baltic Sea and then create a surge on top of it. Our results show that, on average, the opposite is the case, making these 'outliers' stand out even more. Historical ESL events, such as the 2005 (Suursaar and Sooäär, 2007; Mäll et al., 2017) or the 1872 surge (e.g. Hofstede and Hamann, 2022; Andrée et al.,

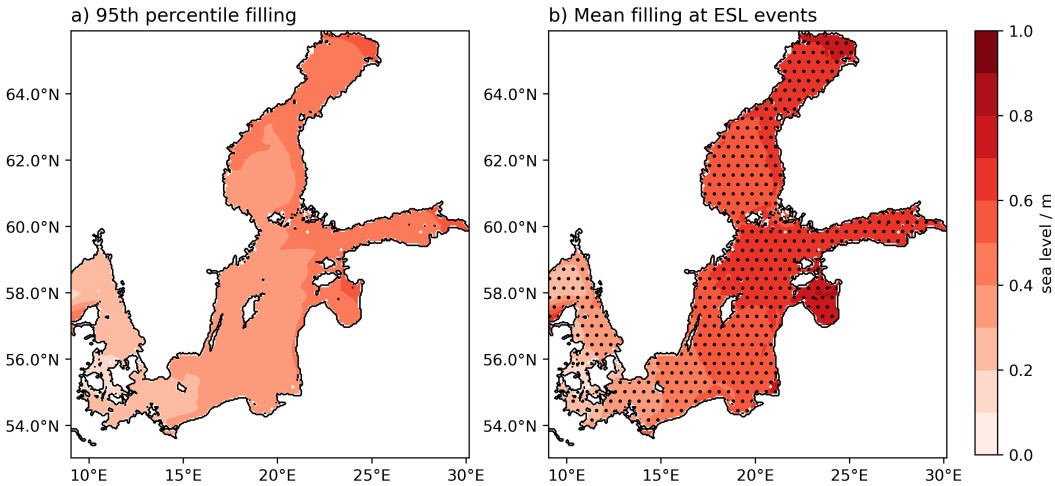

**Figure 10.** Comparison of the 95th percentile of the filling component (a) to the mean filling component during the ESL events (b). The hashed area indicates, where the mean filling during the ESL events is higher than the 95th percentile in a).

2023), were a superposition of extremely high filling (71 cm at Landsort, up to 100 cm in the Western Baltic Sea for 1872 and >70 cm for 2005) and an exceptional surge. MacPherson et al. (2023) argue for the inclusion of these historical events, as they substantially influence (increase) the design sea levels. This is particularly important for the Western Baltic Sea, where the ESL activity has been relatively low over the last 70 years, which is the basis for current ESL statistics and design levels (MacPherson et al., 2023).

Note that none of the statistical approaches mentioned have considered the temporal decomposition of the low and high frequency contributions. If both the filling and the high frequency (surge + seiche) components were treated separately, e.g. a separate GPD distribution for the filling and the high-frequency signal, the return levels would be greatly underestimated since the occurrence of high filling levels and surges is not independent. Comparing the 95th percentile of the filling component, which could serve as a peak-over-threshold for GPD statistics, with the mean filling during the ESL events considered in

this study (Fig. 10) highlights a key point: the mean filling during ESL events consistently exceeds this percentile threshold across the entire Baltic Sea. This demonstrates a clear coupling between high filling levels and surges, suggesting that a combined statistical distribution, as is traditionally used, remains preferable. However, as discussed in the previous paragraph, this approach may not accurately reflect historical events.

Incorporating the correlations between the three temporal components using copula statistics could enhance the statistical

representation of ESLs in the Baltic Sea. This method may help reducing uncertainties in the tail ends of ESL distributions, which are a major source of error when determining design levels for coastal protection. To the best of our knowledge, this approach has not yet been applied to ESLs in the Baltic Sea region.





### 4.3 Consequences for numerical modelling and climate projections

The clear and dominant importance of winds for the generation of ESLs in the Baltic Sea is expected from physics but poses
a problem for numerical ocean models. Since the uncertainty of the storm systems in the atmospheric data is related to the
uncertainties in the representation of ESLs in the ocean models (Lorenz and Gräwe, 2023a), the models depend on good
atmospheric data.

This poses a significant challenge in estimating how ESLs will respond to climate change in the region. Hindcast simula-
tions demonstrate a wide variation in ESL statistics, even when using an ensemble approach (Lorenz and Gräwe, 2023a). The
uncertainty is further compounded by the unpredictable evolution of storm tracks and storm intensities, making future projec-
tion even more uncertain (see the recent Baltic Earth Assessment Reports and references therein, e.g. Meier et al., 2022a, b;
Rutgersson et al., 2022; Weisse et al., 2021). Therefore, the current state of knowledge is that the storm statistics will most
likely stay the same in the future Meier et al. (2022b), since there is no robust change in storms in current regional climate
models for the Baltic Sea region. However, it is certain that sea ice cover will decrease in the future and the mean sea level will
rise (Fox-Kemper et al., 2021; Gräwe et al., 2019). The latter will raise the extreme value statistics to higher baseline levels
(Wahl et al., 2017; Hieronymus et al., 2018). For shallow lagoon systems, such as those in the western and southern Baltic
Sea, sea-level rise will alter water exchange with the Baltic Sea, most likely increasing ESLs there (Lorenz et al., 2023). For
the former, increasing ESLs would be expected where sea ice is currently present, e.g. in the Gulf of Bothnia and the Gulf of
Finland.

Since future ESL statistics are likely to stay the same, except for higher base levels due to sea level rise and the retreat of sea
ice cover, future efforts may be directed towards a better understanding of the current distribution and dynamics of ESLs in the
Baltic Sea region. This statement applies to physics, e.g. interactions of the different low and high-frequency components and
their statistical dependencies.

### 5 Summary and Conclusions

We decomposed observed and numerically modelled extreme sea level (ESL) events in the Baltic Sea into three distinct tem-
poral components: the low-frequency filling of the basin (preconditioning), storm surges, and basin-wide inertial oscillations,
known as seiches. Our findings indicate that the surge component is the dominant contributor in the Western Baltic Sea, while
the filling component plays the primary role in the central and northern parts of the basin. Although seiches contribute the least,
they remain a non-negligible factor.

Through numerical simulations, we further examined the driving forces behind each temporal component. Wind emerged as
the primary driver for all components, followed by atmospheric pressure. Atlantic sea levels significantly influence the surge
component in the Kattegat and Skagerrak regions, located outside the Baltic Sea. However, the Atlantic sea level primarily
affects the filling component within the Baltic Sea. Sea ice was found to impact only the surge component in the northern Baltic
Sea, while interannual variations in sea ice, salinity, and temperature had no substantial influence on any of the components.



The decomposition framework introduced in this study provides a novel and practical approach for isolating the temporal components and their respective driving forces contributing to ESL events in a non-tidal, semi-enclosed basin like the Baltic Sea. Depending on data availability, this method can be easily adapted for other regions, incorporating additional factors like tides in tidal systems or wave setup. Our framework is valuable for advancing the global understanding of sea level extremes and enhancing coastal risk management strategies.

*Code and data availability.*    This study uses the model code and setup configuration of Lorenz and Gräwe (2023a) which can be accessed here: https://doi.org/10.5281/zenodo.8340649 Lorenz and Gräwe (2023b). Additional model simulations for the forcing decomposition are stored here: https://doi.org/10.5281/zenodo.13903910 (Lorenz and Gräwe, 2024).

*Author contributions.*    ML and UG designed the study together. UG created the model chain. ML performed the simulations and the analyses. ML wrote the first draft of the manuscript with input from KV and UG. All authors contributed to the discussion and conclusions.

*Competing interests.*    The authors indicate no competing interests.

*Acknowledgements.*    ML was supported by the Collaborative Research Centre TRR 181 "Energy Transfers in Atmosphere and Ocean" (project 274762653), funded by the German Research Foundation (DFG). All simulations were performed on clusters of the German National High Performance Computing Alliance (NHR). Figures in this paper use the colourmaps of the cmocean package (Thyng et al., 2016).



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
