# Peer review of "Untangling the Waves: Decomposing Extreme Sea Levels in a non-tidal basin, the Baltic Sea"

_Natural Hazards and Earth System Sciences, 2024_

## Author Comment (AC1)

**Reply to Mika Rantanen**

Marvin Lorenz, Katri Viigand and Ulf Gräwe

January 6, 2025

We are grateful for your constructive and in-depth review of our manuscript which helped us to clarify the presentation and interpretation of our results. Our responses to your comments are in blue text below.

**1 Responses**

General comments

This manuscript provides quantitative decomposition of extreme sea level (ESL) events in the Baltic Sea into storm surges, filling and seiches. These components are further decomposed into components from various forcings such as wind and atmospheric pressure. The authors use sea level observations from the entire Baltic Sea coastline and simulations with a numerical model. One of the key results is that storm surges dominate ESL events in the western Baltic Sea, while the filling contribution is more important in the central and northern sea regions.

I found the topic of this manuscript highly relevant and valuable. Here in Finland, it is often stated that surges, filling (preconditioning), and seiches together cause the highest sea levels in the Baltic Sea. However, concrete evidence quantifying their relative contributions has been lacking. The same applies to the specific roles of wind and atmospheric pressure in driving high sea levels. In my opinion, this study fills an important gap by providing detailed quantification of these processes.

I think the overall presentation of the manuscript and its language was very good. The structure was logical, and there was a "red line"

in the story which made the reading enjoyable. The background was nicely covered with relevant references, making the impression that the authors do know the topic very well. The results were discussed in a detailed way from various perspectives at the end of the paper. While I am not a marine scientist, I still understood most of the text.

Despite the positive feeling I got from the reading, I still found a few aspects which in my opinion require clarification: these are related to 1) methods and 2) the negative correlation between storm surges and filling. These are explained below. In any case, I can recommend publication after these (minor) comments have been addressed.

Thank you very much for the positive feedback.

Minor comments

In section 2, you present the observations (2.1), the model simulations (2.2) and the diagnostic decomposition method (2.3). There were some parts which I think were missing:

In 2.3.1 (L115) you do not explain whether the decomposition is done for observations or model simulations or both. When I first did read these sections I assumed that the decomposition was only done for simulated sea level heights, so it was surprising to see in Fig. 3 that the method is applied to both. There is a brief mention in L92 that the decomposition is also done for observations, but this point should be emphasised later when presenting the decomposition method.

We now explicitly mention at the end of Section 2.3.1 that the decomposition of the observed ESLs is performed, and this is further explained in the first sentence of Section 3.1 to clarify that the temporal decomposition is first done for the observations. As a second step, we compare the model's decomposition and show that the model can replicate the statistics. Section 2.3.1: "This temporal decomposition is done for both the observed and modelled ESLs." and section 3.1.: "The temporal decomposition of the observed ESLs [...]"

At Section 2.3.1, you could explain how you derive $\eta_{ESL}$. Is that extracted from modelled data or directly from observations?

Added: "ESL events $\eta_{\mathrm{ESL}}$ are identified by applying a peak-over-threshold method. The threshold is determined by the 99.7th percentile of the time

series, and we only consider events that are separated by morethan 48 hours (Arns et al., 2013)."

> At L95, it would be clearer to write that the observational time series for different tide gauges are of different lengths. Or are they? And I assume that the detrending is based on the linear trend of the whole time series and not on a fixed period.

Yes, the time series are of different lengths. However, we only consider the time span which is also covered by the model run, so 1961 to 2018 in our case, where many gauges are of shorter lengths. We added these information to the section: "Before any analysis, we selected the years 1961 to 2018 (the same time period covered by the model run, see next section) and subtracted the long-term linear trend of the mean sea level for each tide gauge. De-trending removes mean sea level rise and glacial isostatic adjustments (GIA Peltier, 2004) from the time series. Note that some gauges have records shorter than the considered time period and many contain temporal gaps (Lorenz and Gräwe, 2023)."

> In Section 2.2, it would make the choice of the model more robust if you could briefly mention whether the model has been used successfully in some previous studies.

The model has been used for over 2 decades and has been applied in several published hydrodynamic studies including extreme sea levels in the past. To not inflate the references, we kept the literature to previously cited studies on the Baltic Sea, although GETM has also been applied to several estuaries and marginal seas around the globe. "GETM has been used to model the hydrodynamics of several marginal seas and estuaries around the world. It has successfully demonstrated its ability to capture the complex hydrodynamics of the Baltic Sea (Gräwe et al., 2015), the mean sea level dynamics of the Baltic Sea (Gräwe et al., 2019), and its extreme sea levels (Lorenz and Gräwe, 2023; Kiesel et al., 2023)."

> In Section 2.2. It was not clear for me what was the temporal resolution (hourly?) of the simulations, and how long were the simulations? And did you simulate the whole year, including the summer season? Overall the time period which was studied should be written more clearly (I found it from L147 but it could come earlier).

We simulated the whole period from 1961 to 2018. We added the requested information plus additional information on the resolution of the atmospheric forcing: "The spatial resolution of the UERRA forcing is 11 km and the temporal resolution is hourly." and "The simulation period is 1961 to 2018. We save sea level data in a time step of 20 minutes."

> In Section 2.4 (L158-163), the method of calculating the relative contributions of the forcings for sea level remained a little unclear to me. Could it be demonstrated using a single station example? Like writing down the magnitudes of the relative contributions from a station in Fig. 2d. This came back to me when I tried to interpret the sea ice contribution from Fig 7. You say (probably correctly) that its contribution is negative, but in Fig 7 they all look positive because they are presented as pie charts. Is there a contradiction, or have I misunderstood?

We now include a list of relative contributions for the example in Fig. 2d in the text: "For this station, the mean temporal contributions are: 60.5% surge, 26.5% filling, and 12.9% seiche (sum: 99.9%)." and "For the station Warnemuende, the mean forcing contributions are: 84.5% wind, 8.0% air pressure gradients, 2.9% Atlantic sea level, 0.2% baroclinicity of seawater, -0.5% sea ice, 0.2% sea ice variability, and 1.8% residual (sum 97.1%). Due to the uncertainty in the fits, we cannot expect the sum of all forcings to add to 100.0%." For the pie charts, only the absolute values are considered since negative pie pieces are graphically not representable. The contribution of sea ice is indeed negative on average. Sorry for the confusion. We are mentioning this now in the captions of Fig. 7-9. "Note that the pie chart depiction only considers the absolute values and not the sign."

> At Section 3.1.1 (L201-213), I didn't really understand the reason why filling and surges are negatively correlated, especially because in Fig. 2b they seem to be positively correlated (both are positive at the time of maximum). I read several times the sentence "Since the peak sea level of each event is fixed, a particularly high surge would naturally coincide with a lower filling state relative to the mean of the Gaussian distribution.", but I still didn't get the idea.

If one would consider the whole time series of the filling and the surge components, then the correlation could indeed be positive as Fig. 2b suggests

because high filling (compared to mean filling sea level of zero) usually occurs during ESLs. Figure 5 indeed shows that all filling levels are well above zero for all considered events. However, we correlated the statistical samples that make up the Gaussian distributions in Fig. 2d. The Pearson's correlation coefficient compares the samples to their respective means. In our case, the "means" are the mean values from the respective Gaussian distributions. Since the surge and filling are making up most of the relative peak sea level ($\sim 60\%$ surge and $\sim 25\%$ filling, Fig. 2d), both components must be negatively correlated, as otherwise the average relative peak level would be above 100% which would indicate an error in the decomposition. We added more information to the paragraph which hopefully makes our argumentation more clear: "The negative correlation between surge and filling is surprising at first sight, as it is known that most ESLs in the Baltic Sea are a coincidence of both contributions. At second glance, it does not contradict the fact that most ESLs are a combination of high filling states and storm surges which would intuitively indicate a positive correlation. The negative correlation between the relative filling and surge components is partly an artefact of the decomposition method since we do not correlate the whole time series, but only the extracted statistical samples which we used for the Gaussian distribution fits. Since the peak sea level of each event is fixed, a particularly high surge (right side of the surge's Gaussian distribution) must naturally coincide with a lower filling state relative to the mean of the Gaussian distribution (left side of the filling's Gaussian distribution), which emphasizes a negative correlation. Otherwise, the average relative ESL would be higher than 100% and thus the fitted Gaussian distributions would be wrong. The same applies to a low surge and a high filling state. Therefore, these two components must be negatively correlated in this decomposition approach of the mean ESL."

From a meteorological perspective, strong cyclones are typically associated with (long-lasting) westerly winds, which would intuitively lead to a positive correlation between storm surges and filling. Given that this result appears to be one of the key findings of the study, and also being in an apparent contradiction with other studies, I suggest clarifying the mechanism in greater detail. Providing additional explanation would help resolve this apparent contradiction and strengthen the manuscript's conclusions.

This point is based on the misunderstanding of the sample that went into the correlation analysis from the previous comment. Our results are no contradiction to the mentioned mechanism and the co-occurrence of high filling and high surges which intuitively indicate a positive correlation, but only if the means would be zero which is not the case for our samples. Since this is a methodological artefact, we cannot provide details to a mechanism.

Other, specific comments
L12: This phenomenon: does this refer to the rising mean sea level or ESL events? Isn't the ESL events the main cause of flooding, with a smaller contribution from rising sea level?

*Phenomenon* refers here to the ESLs. Due to the rising mean sea level, critical sea levels are nowadays already reached with smaller surges than before.

L31. By input data you mean weather prediction models or reanalysis? Can you mention them explicitly as I was wondering what input data is specifically meant here.

Here we refer to long-term reanalyses data. Rephrased to: "However, long-term, high-quality, high-resolution reanalysis data that meet these requirements are often not available."

L62 These three temporal. Would it be better to put the three components together in brackets, for example, so that the reader does not have to go back to the previous page to see what the three were?

Added the components in brackets: "Together with *storm surges*, these three temporal components (filling, seiches and storm surges) are the main contributors to ESL events."

Table 1. TSClim: temperature or salinity?

The 'of' is an error and should be an 'and'. We rephrased to: "Simulation where the inter-annual variability of temperature and salinity are excluded by using a climatology of temperature and salinity fields."

Table 1. IceClim: inter-annual is written twice. And what does it mean by neglecting the inter-annual variability? Do you run the model with climatological sea ice cover?

Fixed the word doubling. Yes, this simulation is using an ice climatology, i.e. for each simulation year, the same sea ice is present. Similarly, for the TSClim simulation, we use a climatology of T and S, see previous comment for the rephrased description. We rephrased the description of the IceClim simulation: "Simulation where the inter-annual variability of sea ice cover is excluded by using a climatology of sea ice cover."

L109. Does this mean you performed seven 58-year simulations?

Yes. We added the information: "We exclude different forcing components in distinguished model simulations to study the effects of different forcings on the sea level, see Tab. 1, i.e. seven simulations of the period 1961 to 2018."

L237. .. up to 30 %. This sentence remains a bit incomplete. Where does it contribute and what? Can you rephrase it?

Rephrased the sentence to: "However, the Atlantic sea level component can contribute up to 30% of the surge levels for this region."

L252 and L254 I think you write two times the residual term contribution? Is the 2nd (40%) for Danish Straits?

You are right. We meant the Skagerrak and Kattegat and not the Baltic Sea. Rephrased to: "The wind and Atlantic sea level forcings explain most of the filling for the Skagerrak and Kattegat. However, the residual term shows values up to 40% across these regions."

L258. As a meteorologist, I thought first that baroclinicity means atmospheric baroclinicity. Could it be rephrased to add seawater here?

changed to: "[...] baroclinicity of seawater [...]"

Figure 7-9. Related to minor comment 1f. I don't understand how the negative contributions from e.g. sea ice forcing is presented in these pie charts. For me it looks like all the forcings are contributing positively.

Yes you are right. For the pie charts only the absolute values are shown since negative values are not representable in such depiction. We added this information to the captions, see also a previous comment above.

L271 wind systems. Maybe wind climatology is a better term here?

Changed to "wind climatology".

L279 its mean → the mean contribution of filling

Changed.

L281 on this time scale. Which time scale?

We mean that the water exchange with with the North Sea occurs on the same time scale as the time scale of the filling, i.e. ∼ 7 days. Rephrased to: "This is partly because water can flow into the Kattegat, Skagerrak and the North Sea on the time scale of the filling, which is about a week."

L288 Do you speak about the potential increase due to seiches here? It could be added to the sentence.

We speak about the increase by both filling and seiches, although seiches being the main contribution. Rephrased to: "Nevertheless, the potential average increase due to temporal shifts in the seiche and filling is in the order of 10-20 cm."

L296: 10% on average. Was this result shown in some figure? If not, better to add "not shown".

Yes, Fig. 7b shows that sea ice can reduce the surge component for the northernmost stations by almost 10%.

L303 ... currently very small. Maybe add reference to Figure?

Added reference to Figs. 7-9: "The same arguments can be made for the inter-annual variability of sea ice. Our results show that this contribution to the ESLs is currently very small (Figs. 7-9)"

L339. Aren't meteotsunamis more of a summer phenomenon, so that they generally don't occur at the same time as wind-driven extreme sea level events, which tend to occur in the winter season? If this is the case, it could be mentioned here.

In principle, the occurrence of meteotsunamis should not be limited to summer. Pellikka et al. (2022) differentiate between summer and winter type events of meteotsunamis, at least for the Northern Baltic Sea. As there is evidence that meteotsunamis in principle can occur at the same time as a surge, e.g. Pattiaratchi and Wijeratne (2015), we rephrased to: "Nevertheless, meteotsunamis could occur during an ESL event (Pattiaratchi and Wijeratne, 2015) and are a common phenomenon in the Baltic Sea (Pellikka et al., 2020, 2022) with high sea level contributions in the order of decimetres."

**References**

Arns, A., Wahl, T., Haigh, I., Jensen, J., and Pattiaratchi, C.: Estimating extreme water level probabilities: A comparison of the direct methods and recommendations for best practise, Coastal Engineering, 81, 51–66, https://doi.org/10.1016/j.coastaleng.2013.07.003, 2013.

Gräwe, U., Naumann, M., Mohrholz, V., and Burchard, H.: Anatomizing one of the largest saltwater inflows into the Baltic Sea in December 2014, Journal of Geophysical Research: Oceans, 120, 7676–7697, https://doi.org/10.1002/2015JC011269, 2015.

Gräwe, U., Klingbeil, K., Kelln, J., and Dangendorf, S.: Decomposing mean sea level rise in a semi-enclosed basin, the Baltic Sea, Journal of Climate, 32, 3089–3108, https://doi.org/10.1175/jcli-d-18-0174.1, 2019.

Kiesel, J., Honsel, L. E., Lorenz, M., Gräwe, U., and Vafeidis, A. T.: Raising dikes and managed realignment may be insufficient for maintaining current flood risk along the German Baltic Sea coast, Communications Earth & Environment, 4, 433, https://doi.org/10.1038/s43247-023-01100-0, 2023.

Lorenz, M. and Gräwe, U.: Uncertainties and discrepancies in the representation of recent storm surges in a non-tidal semi-enclosed basin:

a hindcast ensemble for the Baltic Sea, Ocean Science, 19, 1753–1771, https://doi.org/10.5194/os-19-1753-2023, 2023.

Pattiaratchi, C. and Wijeratne, E. M. S.: Observations of meteorological tsunamis along the south-west Australian coast, pp. 281–303, Springer International Publishing, Cham, https://doi.org/10.1007/978-3-319-12712-5_16, 2015.

Pellikka, H., Laurila, T. K., Boman, H., Karjalainen, A., Björkqvist, J.-V., and Kahma, K. K.: Meteotsunami occurrence in the Gulf of Finland over the past century, Natural Hazards and Earth System Sciences, 20, 2535–2546, https://doi.org/10.5194/nhess-20-2535-2020, 2020.

Pellikka, H., Šepić, J., Lehtonen, I., and Vilibić, I.: Meteotsunamis in the northern Baltic Sea and their relation to synoptic patterns, Weather and Climate Extremes, 38, 100 527, https://doi.org/10.1016/j.wace.2022.100527, 2022.

Peltier, W. R.: Global glacial isostasy and the surface of the ice-age Earth: the ICE-5G (VM2) model and GRACE, Annual Review of Earth and Planetary Sciences, 32, 111–149, https://doi.org/10.1146/annurev.earth.32.082503.144359, 2004.

---

## Author Comment (AC2)

**Reply to reviewer 2**

Marvin Lorenz, Katri Viigand and Ulf Gräwe

January 6, 2025

We are grateful for your constructive and in-depth review of our manuscript which helped us to clarify the presentation and interpretation of our results. Our responses to your comments are in blue text below.

**1 Responses**

Overall, this paper leverages a validated model to analyse the components of extreme sea levels along the Baltic Sea, exploring their interactions and relative importance. The study introduces an innovative and engaging approach, and while the main conclusions are not entirely novel, they are well-structured, generalizable, and effectively capture the complexities of the Baltic Sea system. The paper is well-written and easy to follow, making it accessible to a broad audience. I recommend the paper for publication, subject to minor or moderate revisions. Below, I provide some specific comments to further enhance the quality of this already good work. The paper is well written, however, there are some typos here and there. Those that I noticed are mentioned, but I suggest the Authors re-read the document looking for minor typos.

Thank you for your positive comments.

Line 35: The claim about second-order effects I believe should be corroborated a bit more?

Including non-linear interactions in the decomposition is complicated, yet can make a significant contribution to the peak sea levels, e.g. tide-surge

interaction (Idier et al., 2019; Arns et al., 2020). Depending of the time series decomposition approach, the non-linear effects are actually automatically attributed to one of the linear contributions or to a residual term since tide gauge data of course include these effects. We have changed the wording to avoid "second order". We have rephrased the sentence to: "Interactions between the different contributors are often attributed to a residual term."

Table 1: Check typo.

Fixed.

Lines 104-105: Can you explain the reason of the 7% increased wind without only relying on the reference?

The calibration of the model showed that this increase was necessary to avoid a negative bias in the Western Baltic Sea which we already write in the text. That there is a bias in the first place in this region was partly attributed to the 'coarse' resolution of atmospheric models which do not resolve the orography of the Western Baltic Sea correctly since the resolution of approx. 10-20 km would cover the Western Baltic Sea with only a few grid cells (Lorenz and Gräwe, 2023). In our previous study (Lorenz and Gräwe, 2023) we found a negative bias in ESL heights for all of the simulations which have used six different atmospheric reanalyses as forcings. We believe that adding more information to this reasoning to the manuscript would be distracting and it would not the affect the results of relative contributions. Curious readers can always check the reference which is an open access publication in Ocean Science.

Line 118: can you please specify the filter order? Any specific reason why you used Butterworth filter? Can you explain the physics behind 7 days?

The specific order of filtering is presented in section 2.4. To avoid repetition, we decided to avoid mentioning the filter order in this section. There is no specific reason to use a Butterworth filter. Any other filter would work as well. Regarding the physics, 7 days is the main time scale which includes both the mean filling state of the whole Baltic Sea, but also local filling aspects, e.g. due to prevailing winds. Furthermore, for comparability with previous studies (e.g. Soomere and Pindsoo, 2016; Pindsoo and Soomere,

2020) we have chosen a similar time scale. We added some information on the reasoning of 7 days: "The filling state $\eta_{\text{fill}}$ is computed by applying a Butterworth low-pass filter with a cut-off frequency of 7 days, corresponding to the weekly timescale described by Soomere and Pindsoo (2016) and Pindsoo and Soomere (2020). This time scale includes both the average filling state of the whole Baltic Sea and local filling due to persistent winds such as storm systems."

> Line 120: ". . . a time window of +- 7 days" means a window of 14 days centred on the peak? Can you please clarify the overall approach that you used? What are the steps? Can you explain the link with the peak sea level?

Again, we refer here to section 2.4. which includes all the requested details.

> Eq 2: can you please show if the results of the fitting provide realistic amplitude and phases?

Fig. 2b of the manuscript shows an example of the fitting of the seiches. The amplitudes are certainly realistic. The phases are also reasonable. A better illustration that this approach works well, is shown in Fig. 1 below for the station Kemi in the Gulf of Bothnia in the north of the Baltic Sea. For the exemplary ESL event (panel b), the seiche signal is very large and the fitting approach is capturing the seiches really well, which gives us confidence that our approach is well suitable.

> Line 125-126: please clarify and justify your belief about the error. I do think that is negligible in your work, but it would be nice to have some solid ground to say so.

We have added more details and arguments to our justification why the consideration of more frequencies is unnecessary: "Note that even more frequencies could have been added, but the amplitudes of these frequencies are much smaller. In addition, since we are considering $\mathcal{O}(100)$ events per station, the uncertainty estimates based on these statistical samples will be larger than the missing frequency contributions. Therefore, the errors introduced by neglecting these additional frequencies are expected to be negligible for this study."

[Figure]

Figure 1: Exemplary results of the workflow for the decomposition of ESL events as in Fig. 2 of the manuscript, but for the station *Kemi (51)*.

Chapter 2.3.2: how do you justify your decompositions? I am referring in particular to the seiches component.

Seiches are inertial surface waves which are the "back-and-forth sloshing" of sea level perturbations. These perturbations can have multiple origins, but most prominent ones are from wind and air pressure. For example, a seiche can be the inertial response after a storm surge. Therefore, our decomposition into forcings should capture the origins of the perturbations. E.g. by switching off winds, the all seiches triggered by wind will not be present in the simulation, which should reduce the seiches' amplitudes. Of course we cannot exclude that the forced fit approach may fit higher amplitudes. But due to the large number of events, and the clear results of the seiche decomposition (Fig. 8), we have high confidence that our decomposition approach is justified.

Line 143: what did you do for detrending the time series? Can you please show both time series and what you removed to get the detrended one?

We have de-trended the time series by a linear regression over the whole time series which we mentioned in section 2.1. We added the information to the first sentence of section 2.4: "As a first step, the long-term changes in the mean sea level are removed from the time series by de-trending by subtracting the linear trend of the entire time series; see Fig. 2a." You can find the comparison of the original and de-trended time series below in Fig. 2. The de-trending not only subtracts the mean sea level rise over the time period, but also possible biases in the reference heights in the tide gauge observations. We decided to not explicitly show the de-trending in Fig. 2 of the manuscript because this approach of de-trending is straightforward.

Line 148: can you please rewrite the following part "....which are shown in Fig. 1, see Fig. 2a for the identified events." It is not clear.

Rephrased to: "For the modelled time series spanning 58 years from 1961 to 2018, approximately 80 to 250 events are identified for each station. The station locations are shown in Fig. 1."

Line 154: Please clarify the content of the bracket.

[Figure]

Figure 2: De-trending of the sea level data by linear fit over the time series, here the tide gauge data of Warnemuende as an example. For this example, the trend is 1.8 mm/yr, thus a mean sea level increase of ∼ 10.4cm from 1961 to 2018.

Now reads as "An exemplary temporal decomposition of the observed Warnemuende time series (Fig. 1, station 9) is shown in Fig. 2b."

Line 156: it is not clear how you used the maximum values within +-24hours, can you clarify?

Rephrased to "In addition, the maximum filling and maximum seiche level values within a 48-hour window, i.e. ±24 h around the ESL peak, are stored to assess the potential ESL if all three components were to reach their peaks simultaneously."

Line 158: Why do you need to normalise the components? It is not clear this step, please clarify what you did and provide the reasons to do so. Why did you not use the distributions from the original dataset?

We normalise the components to study the respective relative contribution of the different components to the ESL. Normalisation ensures that the different events are comparable to each other which allows us to compute a mean composition of ESLs. We added more information and justification to the approach: "To study the relative importance of each component, the

components are normalised to the peak level for each event and stored in a histogram with a bin size of 0.01, ranging from -0.5 to 1.2. By the normalisation, we can use the all events to make general statement of the mean composition of ESLs."

Line 173: if you use the countries to explain the figure 3.a, please add the borders in the figure.

We added the borders and country names in Figs. 3, 7, 8, and 9.

Lines 187-188: due to the layout it is not clear the resolution you are referring.

We don't have influence on the layout of the text. For type-setting we will keep an eye open for formatting issues like the line break you mention. We now explicitly mention the model to make the sentence more clear : "Differences are found for gauges that are located within coastal lagoons or estuaries, such as Ueckermuende, Althagen, Barhoeft, Kappeln, Schleswig, and Gdansk, since the hydrodynamics are not resolved at the 1 N.M. resolution of the model, as discussed by Lorenz and Gräwe (2023)."

Lines 191-195: explain more clearly the reasoning behind your claims. Moreover, please provide at least some names of the locations on the map too. Moreover, Cuxhaven is not in the Baltic sea, why do you present?

We included station numbers instead of names and elaborate more on the reasoning. The Cuxhaven station is included as a contrasting station where different dynamics should be present. We rephrased the paragraph to: "Although located outside the semi-enclosed Baltic Sea, stations in the Kattegat and Skagerrak (stations 20-28) and in the North Sea (station 73) still exhibit large filling contributions of 40-50%. This shows that, at least for ESL events, low-frequency waves contribute to the slow sea level variability. This makes sense as the mean filling of the Baltic Sea is controlled by the water exchange with the North Sea, which requires long periods of elevated sea level in front of the Danish Straits. Since tides are excluded in the simulations, the low-frequency variability cannot be a superposition effect such as the spring-neap cycle, further indicating that low-frequency waves also contribute significantly to ESLs in the more open eastern North Sea. As a

contrast to the Baltic Sea stations, we included one station in the North Sea, Cuxhaven (72). The model deviates from the observations for this station in the southeastern North Sea because tides are not included in the model. Nevertheless, the filling contribution of about 25% to the ESLs is a noteworthy result, indicating that persistent westerlies can elevate the mean sea level for a period of at least one week and longer in this region."

> Chapter 3.1.1: As you mentioned, the selection method (i.e. POT) and the assumed linear summation together induce at least some of the mentioned negative correlation. Can you please explain why this is less important in the surge/seiches correlation and why you keep having a positive correlation? Your explanation of the positive correlation is ok, but why the induced negative part is here less important?

We now discuss the low, but significant, negative correlation between the seiche and the filling: "The negative correlation between filling and seiches shows smaller coefficients than between filling and surge components, indicating that seiches tend to be small when filling is high and vice versa. Since there is a positive correlation between surges and seiches, and a negative correlation between filling and surges in the same areas, it makes sense that seiches should generally be negatively correlated with filling as well."

> Can you please provide the details of the correlation analysis? Which correlation coefficient did you use? Can you please mask the map points having the p-values lower than a reasonable threshold? Without any diagnostic checks, the map can be misleading.

Thank you for the suggestion. We use the Pearsons's correlation coefficient which we now mention in the text. We now marked areas where the p-value is below 0.05, i.e. statistical significance by hatches, see Fig. 3 below for the new version of the plot. This is a very good addition to support our argumentation.

> Chapter 3.1.2: I am unsure whether it is needed for the paper. I do not require the chapter to be removed, I leave this to the Authors, but I believe it's not so interesting as the rest of the document.

Thank you for the suggestion. While this chapter may not be as interesting as the other chapters, we believe that the quantification of potential maximum

[Figure]

Figure 3: Correlation maps for the three temporal components: a) Correlation coefficient for *surge* and *filling* components. b) Correlation coefficient for *filling* and *seiche* components. c) Correlation coefficient for *seiche* and *surge* components. The hashed areas indicate where the p-value is below 0.05.

sea level increases to be useful information since it illustrates the partial randomness of the phases of the three contribution to each other. Furthermore, it provides an indication that it does not necessarily require climate change to significantly increase ESLs which experts of course know, but non-experts may overlook.

> Figures 3 and 7: an option to have an idea of the location would be to add to the name of the locations on the x axis a number and reproduce these numbers in the map. In any case, something to help the localisation of the different locations on the maps should be done. I leave it to the author's preferences how. Please increase the y axis of the bar plot series.

We added the numbers of the stations to the x-labels. However, we do not add the numbers to the pie charts to avoid making the plots too messy. Fig. 1 shows the locations of the gauges by the numbers which we believe is enough. Already in Fig. 1 our feeling is that the numbers make the map a little messy which would be worse when the pie charts are present in the same plot. We further added country borders and names to each map. We have also stretched the y-axes in panel b) of Figs. 7-9 which enhanced the clarity of the plots.

> Lines 255 and everywhere within the document specify the meaning of low-frequency waves? What are you referring to?

With low-frequency waves, we mean the filling component, which is a surface wave with a long period, thus it has a low-frequency. Since we already always mention "filling" right after the term "low-frequency wave" we believe that we cannot be more precise.

> Line 274: check the typo, "is" is missing in the sentence.

Fixed.

> Lines 284-285: is this temporal shift something realistic?

The delay between the filling and surge should be random due to the different time scales of these components. For the seiches, this is different. Only where the correlation between the surge and seiche are close to zero, there is

a potential of increased ESLs, e.g. everywhere, but in the central Baltic Sea. Therefore, we believe that it is indeed realistic.

Lines 288-290: the sentence seems not finished. If these values do not represent that, then they represent what. . . ?

We clarified the text to: "These values represent the theoretical maxima of ESLs in the respective regions for the specific events."

Lines 292-293: The sentence "This result is expected since most ocean surface waves are forced by momentum transfer from the atmosphere to the ocean by winds or by atmospheric pressure via the inverse barometric effect." might also be removed.

Removed

Lines 299-302: I do not see the link between the following sentences and the paper. I suggest to remove because not relevant for this paper, or otherwise justify the reason to be mentioned. "However, with decreasing levels of sea ice due to climate change (e.g. Meier et al., 2022a, b, and references therein), the contribution of storm surge to ESLs is likely to increase in the future in regions that are currently covered by annual sea ice. In addition, with decreasing ice cover, the average wave loads and annual wave energy flux are expected to increase by about 5% and up to 82% respectively (Najafzadeh et al., 2022)."

We decided to keep the sentence regarding the future decline of sea ice which is likely increasing the relative importance of storm surges to ESLs. However, we removed the sentence regarding the wave loads, since wind waves do not fit to this section.

Chapter 4.1.2: can be removed and for each excluded components specify what is the expected effect(s) on the main outcomes. It is already partially done, but I believe is interesting can be detailed a bit more addressing the effects rather than the reason why each component was not considered.

We do not understand what the "removed part" of the comment refers to. Nevertheless, we agree that we should add more details on the consequences

for our results, especially for meteotsunamis and river discharge. The last two paragraphs now read as: "Similar to wave setup, we have excluded meteotsunamis (Monserrat et al., 2006; Pattiaratchi and Wijeratne, 2015b). These are tsunami-like surface waves generated by the matching of the propagation speeds of a small atmospheric pressure jump and the induced surface wave, e.g. a Proudman resonance (Proudman, 1929). The reasons for neglecting meteotsunamis are simple: First, the temporal and spatial resolution of the meteorological forcing is too coarse to resolve the propagating pressure system accurately enough to generate meteotsunamis in the numerical model. Second, the hourly resolution of the observational data used is also too coarse to resolve meteotsunamis that occur on faster time scales (minutes). Nevertheless, meteotsunamis could occur during an ESL event (Pattiaratchi and Wijeratne, 2015a) and are a common phenomenon in the Baltic Sea (Pellikka et al., 2020, 2022) with high sea level contributions in the order of decimetres. Meteotsunamis could easily be included in the temporal decomposition using a high-pass filter. We have also ignored the influence of river discharge since the coarse resolution of our model does not sufficiently resolve the estuaries and constrictions where river discharge increases sea level. However, compounding ESLs with very high river discharge can elevate the peak sea level (Talke et al., 2021) and have been observed in the southwestern Baltic Sea (Heinrich et al., 2023). We do not expect major changes in our results, as these effects are restricted to estuaries and we have studied ESLs at the open coast. Nevertheless, we acknowledge the importance of river discharge in estimating coastal flooding."

Lines 319-320: "However, the potential contribution of wave setup can be substantial in specific locations." can you be more specific?

We added specific examples: "However, the potential contribution of wave setup can be substantial in specific locations, e.g. for exposed coasts of islands (Su et al., 2024) or coastal bays (Soomere et al., 2013)."

Lines 375-378: if you consider the two statistics completely independent, the final event resulting from the sum of the two components having the same probability of exceedance is larger than accounting for the correlation between the components. Can you please revised the text?

Rephrased the text to: "If the statistics of two (or more) components were

considered independently, the peak sea level resulting from the summation of the sea levels of the components with the same return period (the same probability) would be overestimated, because correlations between the components are neglected."

Line 389: "...which could serve as a peak-over-threshold for GPD statistics,..." I think is redundant and slightly misleading.

Removed.

**References**

Arns, A., Wahl, T., Wolff, C., Vafeidis, A. T., Haigh, I. D., Woodworth, P., Niehüser, S., and Jensen, J.: Non-linear interaction modulates global extreme sea levels, coastal flood exposure, and impacts, Nature Communications, 11, 1–9, https://doi.org/10.1038/s41467-020-15752-5, 2020.

Heinrich, P., Hagemann, S., Weisse, R., Schrum, C., Daewel, U., and Gaslikova, L.: Compound flood events: analysing the joint occurrence of extreme river discharge events and storm surges in northern and central Europe, Natural Hazards and Earth System Sciences, 23, 1967–1985, https://doi.org/10.5194/nhess-23-1967-2023, 2023.

Idier, D., Bertin, X., Thompson, P., and Pickering, M. D.: Interactions between mean sea level, tide, surge, waves and flooding: mechanisms and contributions to sea level variations at the coast, Surveys in Geophysics, 40, 1603–1630, https://doi.org/10.1007/s10712-019-09549-5, 2019.

Lorenz, M. and Gräwe, U.: Uncertainties and discrepancies in the representation of recent storm surges in a non-tidal semi-enclosed basin: a hindcast ensemble for the Baltic Sea, Ocean Science, 19, 1753–1771, https://doi.org/10.5194/os-19-1753-2023, 2023.

Monserrat, S., Vilibić, I., and Rabinovich, A. B.: Meteotsunamis: atmospherically induced destructive ocean waves in the tsunami frequency band, Natural Hazards and Earth System Sciences, 6, 1035–1051, https://doi.org/10.5194/nhess-6-1035-2006, 2006.

Pattiaratchi, C. and Wijeratne, E. M. S.: Observations of meteorological tsunamis along the south-west Australian coast, pp. 281–303, Springer International Publishing, Cham, https://doi.org/10.1007/978-3-319-12712-5_16, 2015a.

Pattiaratchi, C. B. and Wijeratne, E.: Are meteotsunamis an underrated hazard?, Philosophical Transactions of the Royal Society A: Mathematical, Physical and Engineering Sciences, 373, 20140 377, https://doi.org/10.1098/rsta.2014.0377, 2015b.

Pellikka, H., Laurila, T. K., Boman, H., Karjalainen, A., Björkqvist, J.-V., and Kahma, K. K.: Meteotsunami occurrence in the Gulf of Finland over the past century, Natural Hazards and Earth System Sciences, 20, 2535–2546, https://doi.org/10.5194/nhess-20-2535-2020, 2020.

Pellikka, H., Šepić, J., Lehtonen, I., and Vilibić, I.: Meteotsunamis in the northern Baltic Sea and their relation to synoptic patterns, Weather and Climate Extremes, 38, 100 527, https://doi.org/10.1016/j.wace.2022.100527, 2022.

Pindsoo, K. and Soomere, T.: Basin-wide variations in trends in water level maxima in the Baltic Sea, Continental Shelf Research, 193, 104 029, https://doi.org/10.1016/j.csr.2019.104029, 2020.

Proudman, J.: The Effects on the Sea of Changes in Atmospheric Pressure, Geophysical Supplements to the Monthly Notices of the Royal Astronomical Society, 2, 197–209, https://doi.org/10.1111/j.1365-246X.1929.tb05408.x, 1929.

Soomere, T. and Pindsoo, K.: Spatial variability in the trends in extreme storm surges and weekly-scale high water levels in the eastern Baltic Sea, Continental Shelf Research, 115, 53–64, https://doi.org/10.1016/j.csr.2015.12.016, 2016.

Soomere, T., Pindsoo, K., Bishop, S. R., Käärd, A., and Valdmann, A.: Mapping wave set-up near a complex geometric urban coastline, Natural Hazards and Earth System Sciences, 13, 3049–3061, https://doi.org/10.5194/nhess-13-3049-2013, 2013.

Su, J., Murawski, J., Nielsen, J. W., and Madsen, K. S.: Coinciding storm surge and wave setup: A regional assessment of sea

level rise impact, Ocean Engineering, 305, 117 885, https://doi.org/ 10.1016/j.oceaneng.2024.117885, 2024.

Talke, S. A., Familkhalili, R., and Jay, D. A.: The Influence of Channel Deepening on Tides, River Discharge Effects, and Storm Surge, Journal of Geophysical Research: Oceans, 126, e2020JC016 328, https://doi.org/ 10.1029/2020JC016328, e2020JC016328 2020JC016328, 2021.